# Non-Parametrical Canonical Analysis of Quality-Related Characteristics of Eggs of Different Varieties of Native Hens Compared to Laying Lineage

**DOI:** 10.3390/ani9040153

**Published:** 2019-04-09

**Authors:** Antonio González Ariza, Francisco Javier Navas González, Ander Arando Arbulu, José Manuel León Jurado, Cecilio José Barba Capote, María Esperanza Camacho Vallejo

**Affiliations:** 1Department of Genetics, Faculty of Veterinary Sciences, University of Córdoba, 14071 Córdoba, Spain; angoarvet@outlook.es (A.G.A.); anderarando@hotmail.com (A.A.A.); 2Centro Agropecuario Provincial de Córdoba, Diputación Provincial de Córdoba, 14071 Córdoba, Spain; jomalejur@yahoo.es; 3Department of Animal Production, Faculty of Veterinary Sciences, University of Córdoba, 14071 Córdoba, Spain; cjbarba@uco.es; 4Instituto de Investigación y Formación Agraria y Pesquera (IFAPA), Alameda del Obispo, 14004 Córdoba, Spain; mariae.camacho@juntadeandalucia.es

**Keywords:** Egg quality, color coordinate decomposition, internal quality traits, external quality traits, DSM color fan

## Abstract

**Simple Summary:**

The development of new more productive lines of laying hens has displaced native breeds to second place; therefore, new lines of research that ensure the conservation of local breeds and biodiversity are increasingly necessary. The aim of the present study is to characterize the productive capability of Utrerana and to compare the relationships among parameters determining the internal and external quality of the egg, through canonical correlation analysis. We used a flock of 68 Utrerana hens with animals of each of its four varieties (white, black, Franciscan and partridge), and a group of 17 Leghorn hens as a control group. The breed and variety significantly affected egg characteristics. The external and internal quality parameters of the egg were evaluated and reported results providing consistent data for the characterization of the products from this breed. This productive characterization could benefit the conservation of the Utrerana breed, the establishment of livestock models that adapt to it and the search for a market in which this product could be used.

**Abstract:**

The aim of the present study is to characterize the productive capability of Utrerana and to compare the relationships among parameters determining the internal and external quality of the egg, through canonical correlation analysis. A flock of 68 Utrerana hens and a control group of Leghorn hens (n = 17) were housed individually to allow individual identification of eggs and for the assessment of egg quality characteristics. Almost all variables showed differences when both breeds were compared, except for white height, yolk diameter, yolk^L*^ and yolk pH (*p* > 0.05). Only minor diameter, white height, yolk^L*^, yolk^a*^, and shell weight reported significant differences between laying age groups. White height, yolk color, and almost all yolk color coordinates were significantly different (*p* < 0.05) for period and month. Egg and white weight reached highest significantly different levels for the fourth and fifth time that the hens laid an egg. External quality-related traits are better predictors of internal quality-related traits than vice versa, enabling the implementation of an effective noninvasive method for internal quality determination and egg classification aimed at suiting the needs of consumers.

## 1. Introduction

Along with other food such as milk, eggs represent a great contribution of proteins of animal origin to the human diet [1]. In 2017, the world production of eggs exceeded 1416 trillion tons of eggs, equivalent to 80 million metric tons, 30% higher than production in 2000 [2]. In the European Union, the production of food in alternative production systems is on the rise; in 2017, free range and organic egg production accounted for 20% of the total production of eggs [3].

Currently, almost all of the consumed eggs are produced by commercial hybrid lines, which are characterized by high productive performance and a good feed conversion index [4,5]. However, the exploitation of these highly productive lines causes a decrease in the genetic variability of the species and has negative effects on the development of sustainable practices based on local breeds [6].

The emergence of new commercial lines of laying hens with a much greater productive capacity throughout the twentieth century caused the displacement of the autochthonous breed. In many cases, their hybridization with more productive lines relegated autochthonous breeds, including the Utrerana avian breed, to a form of ornamental poultry farming, based on the morphological selection of breeding animals. As a result, a reduction in the census of animals of this breed occurred in addition to a decrease in the productive indices [7].

The Utrerana hen breed was created in the first half of the 20th century, starting with the selection of a heterogeneous population of chickens from the Andalusian countryside [8]. Its initial productive orientation was to be a laying hen, with an annual output of 120–180 eggs, white in color and with an average weight of 62–64 g. It has four different varieties, characterized by the color of the plumage and the legs: white, Franciscan, black, and partridge [9].

The need for the characterization of the products of the Utrerana hen breed is largely due to the situation that it faces. This breed is classified as an endangered breed, according to the Royal Decree Law 2129/26 December, 2008, which establishes the national program of conservation, breeding, and promotion of livestock breeds, and presented a census of 1309 animals on 31 December, 2018 [10]. Therefore, facing this alarming situation, the implementation of programs for the recovery, conservation and productivity improvement of the breed, trying to provide it with an identity and, again, a productive role able to satisfy the demands of the market is required. Thus, the assessment of local products may be a strategy for the conservation of local breeds, for instance, avoiding the loss of linkage between local products and their area of production, as is the case of industrial products [11]. 

Biodiversity must not only be considered as the genetic conservation of animal resources but also the search economic sustainability and the maintenance of the hen population in rural areas [12]. The increasing concern of the society about animal welfare has allowed the development of alternative forms of livestock, including extensive local farms [13]. The Utrerana hen breed, as a local breed, is perfectly suited to this operating system, since it presents great rusticity and resistance to extreme weather situations, with great ability to search for food in the wild [14].

In general, the quality of the egg is related to characteristics that affect the acceptability of eggs by the consumer [15]. Among the considerable number of characteristics of egg quality that can be measured, external factors such as egg weight are the most important [16,17,18,19]. The internal egg quality is also an important aspect to consider, especially when approaching the marketing opportunities of the product. A dense albumen height is among the most important determinants of the internal quality [20,21]. In addition to these factors, other parameters such as the major and minor diameters of the egg, eggshell, yolk color and the weight and pH of the white and yolk allows a more complete characterization of the quality of the egg [22,23,24]. It has been shown that breed genotype can significantly affect most of these features: egg shape, yolk and albumen quality, shell and egg weight and amount of yolk [25]. 

The first objective of this study is to characterize the productive capacity of the four varieties of Utrerana hens compared to a globally distributed laying lineage, as a means of demonstrating the benefits of greater genetic diversity on the quality of products derived from sustainable native breeds. In addition, we quantified the explanatory power of the variance by factors such as the laying month, laying order, period, laying age, variety, and breed found in two sets of parameters of external and internal egg quality. Secondly, we compared the relationships among determining parameters of the internal and external quality of the egg of endangered native hens through a canonical correlation analysis to develop a predictive tool that may enable indirect scoring of the internal quality of the egg from the set of external quality variables.

## 2. Materials and Methods

### 2.1. Animal Sample and Diets

A total of 85 hens were used in the present study, distributed depending on their age and variety as shown in Table 1. The birds were housed in individual cages (50 × 62 × 41 cm) following Council Directive 1999/74/EC of 19 July, 1999, laying down minimum standards for the protection of laying hens at the Centro Agropecuario Provincial de Cordoba (Spain), for 6 months (January to June 2018). All the animals were fed on the same commercial feed (15.2% crude protein, 4.1% calcium, 0.66% available phosphorus) for the whole experimental period. Feed and water were available ad libitum. All the birds were reared according to the regulations of the European Union (2010/63/EU) in their transposition to the Spanish law (RD 53/2013).

### 2.2. Work Sample

All statistical tests were carried out using an egg sample comprising 194 eggs laid from March to June 2018 by the animal sample described above. A total of 147 eggs had been laid by Utrerana hens, while 47 belonged to Leghorn laying hens. The same information registration protocol was followed for all the eggs comprising the sample except for yolk and white pH determination. Due to economic reasons, 97 eggs were chosen at random to perform yolk and white pH analysis.

### 2.3. External and Internal Quality-Related Traits Set Description

Two sets of variables were measured. The first set of variables comprised external quality-related traits, those characteristics that can be measured externally without the need to break the eggs. This first set comprised the variables of egg weight, length and breadth, shell color lightness and shell color coordinates (Shell L*, Shell a*, Shell b*, lightness, red/green and yellow/blue coordinates, respectively). In contrast, the second set of traits, considered internal quality-related variables, required the egg to be broken so as to be scored. This second set comprises albumen height, yolk color, yolk lightness and color coordinate decomposition (YolkL*, Yolka*, Yolkb*), yolk diameter, shell weight, yolk weight, albumen weight, yolk pH and white pH.

### 2.4. Information Registration

Laying lasted for 120 days. All eggs were divided into three periods of 40 days with a mean number of 64.67 eggs per period. Periods ran from second half March to first half April, second half April to first half May, and second half May to first half June. Egg temperature at the time of egg quality assessment was 22 °C ± 1 °C. Individual collection of the eggs of each hen was carried out and all the required variables for the external characterization of the egg were studied daily during the 24 hours following oviposition. Every egg was weighed with a weighing scale (Cobos, CSB-600C, Barcelona, Spain). Major and minor diameters of the egg were measured following a Vernier scale (Electro DH M 60.205, Barcelona, Spain). The color of the shell was determined using a portable spectrophotometer (CM 700d, Konica Minolta Holdings Inc., Tokyo, Japan), and the results were expressed using the International Commission on Illumination (CIE) L*a*b* system color profile (CIE, 1976). 

The traits measured to describe the internal quality of the eggs were as follows: weight of the egg, shell, egg yolk and egg white, white height, the diameter of the yolk, pH of the white and yolk and the color of the yolk. These measurements were taken every fifteen days in all the eggs that the flock of hens laid on the day of collection, evaluating a total of 194 eggs. Then, a sample of 97 eggs was tested at random for yolk and white pH.

To determine internal quality-related traits, the eggshell was broken and the egg contents were deposited on a glass surface. The diameter of the yolk was measured with a Vernier scale. The intensity of the yellow color of the yolk of the egg was measured with the portable spectrophotometer and with a DSM^®^ fan (formerly Roche color fan). The pH of the yolk and white was measured using reactive strips. The height of the white was computed as the mean of three measurements obtained with a Haugh digital micrometer (Baxlo, Barcelona, Spain). Finally, eggshell, egg white, and the yolk were weighed separately using a precision balance.

### 2.5. Statistical Analysis

All variables recorded were separated into two variable sets. The first set included external egg quality-related parameters, such as egg weight, major diameter, minor diameter, shell^L*^, shell^a*^, shell^b*^ and white height, respectively. The second set was internal egg quality-related parameters, such as yolk color, yolk^L*^, yolk^a*^, yolk^b*^, yolk diameter, shell weight, yolk weight, white weight, yolk pH, and white pH.

Levene’s test for equality of error variance was run to test for homoskedasticity. Mauchly’s W Test was run to test for sphericity. All assumptions except for Shapiro Wilk Francia’s normality tests were carried out using SPSS Statistics for Windows, Version 24.0, IBM Corp. (2016). Shapiro Wilk Francia’s normality tests were carried out with the sfrancia routine of StataCorp Stata version 14.2. Skewness and Kurtosis statistics were tested to support the reports by Shapiro Wilk Francia’s normality tests. As the factors (month of laying, laying order, controlled period, laying age, variety, and breed) and variables in the model had violated most of the common parametric assumptions, the decision to follow a non-parametric approach was made. The Mann–Whitney U test was used to compare differences between the two independent groups of the laying age (laying hens and laying pullets) and breed (Utrerana and Leghorn) variables (Appendix A). Similarly, a Kruskal–Wallis H test was performed to study the potentially existing differences between-levels of the same factor when three or more groups existed within the same independent variable (the rest of independent variables) in Table 2 and Appendix A. 

After conducting the Mann–Whitney U test, we assessed the relationship between the factors of laying age and breed and the internal and external quality-related variables tested. Simultaneously, we used the Kruskal–Wallis H test to assess the relationship with the same variables and those factors with three or more categories or groups (k). Then, we computed the strength of the effects of these factors using *r* and partial eta squared (ηp^2^) as quantification measures depending on whether Mann-Whitney U or Kruskal-Wallis H tests had been carried out beforehand (Table 2 and Appendix A). According to Fritz, et al. [28], *r* can be calculated as an effect size for the Mann–Whitney U test using the formula:(1)$$r\;=\;\frac{z}{\sqrt{N}}$$

Cohen’s guidelines [29] for *r* are that a small effect is 0.1, a medium effect is 0.3, and a large effect is 0.5 [30]. Calculation of *r*, *r*^2^, or η^2^ from these z values is possible because
(2)$$r\;=\;\frac{z}{\sqrt{N}}\;\mathrm{and}\;r^{2}\;\mathrm{or}\;\eta^{2}\;=\;z^{2}/N$$


The literature recommends the use of partial eta square instead of classical eta square when using a multifactor design. The reason for this is that, through the use of partial eta square, we report an index of the strength of association between an independent variable and a dependent variable that excludes the variance produced by other variables [31]. The Kruskal–Wallis H test produces chi^2^ values with k − 1 degrees of freedom. We can transform chi^2^ into an F value with k − 1 numerator degrees of freedom (dfn) and N-k denominator degrees of freedom (dfd) using the expression F(dfn,dfd) = chi^2^/(k − 1), modified from Murphy et al. [20]. 

In Cohen’s terminology, a small effect size (0.01) is one in which there is a real effect but which you can only see through careful study. By contrast, a ‘large’ effect size (0.25) is an effect which is big enough, and/or consistent enough that one may be able to see it ‘with the naked eye’.

As almost all the variables have been previously reported to be non-normally distributed (Table 1) (Shapiro-Wilk Francia’s tests (*p* < 0.001), an independent-sample median test was carried out to assess the differences in the median between categories within the same factor (Appendix A). Appendix A shows descriptive statistics for external and internal egg quality-related traits in Utrerana hens compared to the laying lineage in two models, including (n = 97) and excluding (n = 194) yolk and white pH.

Afterward, we studied the pairwise comparisons for any dependent variables for which the Kruskal–Wallis test is significant, aiming at assessing whether there were statistically significant differences between groups of the same factor concerning the external and internal quality-related variables using Dunn’s test (Appendix A). Then to provide a quantifiable measure of such differences, we provide within-group (level) medians in Appendix A. 

We estimated the Pearson product-moment correlation coefficient among variables from both sets using a bivariate procedure from the Correlate package of SPSS Statistics for Windows, Version 24.0, IBM Corp. (2016) [32] to avoid the severe multicollinearity or linear dependency between several variables, aiming at excluding those with multiple correlation coefficients higher than 0.80 according to Montgomery, et al. [33] (Appendix A). Canonical correlation analysis was performed to analyze the relation between the two sets of traits (internal quality and external quality) [34,35]. Therefore, it is possible to define the linear combination of the two sets of variables as [36]:*U*_1_ = *a*_11_*X*_1_ + *a*_12_*X*_2_ + … + *a*_1*p*_*X*_*p*_(3)
*V*_1_ = *b*_11_*Y*_1_ + *b*_12_*Y*_2_ + … + *b*_1*q*_*Y*_*q*_(4)


Canonical variables U_1_ and V_1_ belong to the ith canonical pair associated with the first canonical correlation, expressed for:(5)$$r_{i}\;=\;\frac{côv\left(U_{i},\;V_{i}\right)}{\sqrt{\hat{V}\left(U_{i}\right)\cdot \hat{V}\left(V_{i}\right)}}$$

The percentage of variance explained by the canonical variable $U_{x}^{2}$ and its opposite $V_{y}^{2}$ is determined by: (6)$$U_{x_{i}}^{2}\;=\;\frac{\sum_{j\;=\;1}^{p}a_{ij}^{2}}{p}$$
(7)$$V_{y_{i}}^{2}\;=\;\frac{\sum_{j\;=\;1}^{q}b_{ij}^{2}}{q}$$
where *p* and *q* are the number of variables from X and Y, respectively. To check for the significance of canonical correlation, maximum likelihood ratio test was performed, considering Lambda (Λ) from Wilk’s statistics, following the equations reported in Khattree and Naik [37].

All non-parametric tests were carried out using the independent samples package from the non-parametrical task of SPSS Statistics for Windows, Version 24.0, IBM Corp. (2016). Canonical correlation analysis was carried out using the Canonical correlation procedure from the Correlate package of SPSS Statistics for Windows, Version 24.0, IBM Corp. (2016).

### 2.6. Publication Ethics Statement

All farms included in the study followed specific codes of good practices and, therefore, the animals received humane care in compliance with the national guidelines for the care and use of laboratory and farm animals in research. All subjects gave their informed consent for inclusion before they participated in the study. The study was conducted in accordance with the Declaration of Helsinki. The Spanish Ministry of Economy and Competitivity through the Royal Decree-Law 53/2013 and its credited entity the Ethics Committee of Animal Experimentation from the University of Córdoba permitted the application of the protocols present in this study as cited in the fifth section of its second article, as the animals assessed were used for credited zootechnical use. This national Decree follows the European Union Directive 2010/63/UE, from the 22 September 2010.

## 3. Results

### 3.1. Parametric Nature Assumption Testing

The data was non-normally distributed (Shapiro Wilk’s Francia W, *p* < 0.001) in all cases except for egg weight, major diameter, and white weight. Skewness statistics reported values between −½ and ½, which suggested that almost all variables were approximately symmetric, except for egg weight and white weight which were moderately skewed. All variables presented a distribution with kurtosis <3 (excess kurtosis <0) or platykurtic. Compared to a normal distribution, the central peak of the data distribution is lower and broader, and its tails are shorter and thinner.

Levene’s test for equality of error variance reported that the error variance around the predicted scores was not the same for all the predicted values (*p* < 0.05), except for minor diameter, thus there was no homogeneity of variances for each combination of the levels of the independent variables (species, month, year, and pathology diagnosed); hence, the assumption of homoscedasticity was violated. Mauchly’s W Test of Sphericity (Mauchly’s W = 0.001), χ^2^(104) = 2985.402, *p* < 0.05) indicated that the variances of the differences were not equal; hence, the assumption of sphericity was also violated. 

### 3.2. Factor Variance Explanatory Power and Within Between-Level Differences

The Mann–Whitney U test was used to compare differences between the two independent groups of the laying age (Laying hens and Laying pullets) and breed variables (Utrerana and Leghorn). Almost all variables showed differences when the two breeds were compared, except for white height, yolk diameter, yolk^L*^ and yolk pH. However, only minor diameter, white height, yolk^L*^, yolk^a*^, and shell weight reported a significant difference between the different laying age groups (Table 2 and Appendix A).

The study reports the results from the Kruskal–Wallis H test for all the variables and levels considered in the study and *r* and partial eta squared (ηp^2^) as a measure of the strength of the factors the variables tested (Table 2 and Appendix A). Appendix A show the differences between the median of the categories of the factors; month of laying, laying order, controlled period, laying age, variety, and breed reported by the independent-sample median test. Appendix A show the results for Dunn’s tests pairwise comparisons between the different levels of the factors and variables. 

Dunn’s test pairwise comparisons and the independent-sample median test showed the white variety of the Utrerana hen and Leghorn were not significantly different (*p* > 0.05) for all variables except for egg weight, minor diameter (breadth) and eggshell, with Leghorns reporting the highest median for all the three variables and varieties. The same tests reported a generalized significant difference between eggs from the first lay and the rest of the lays regarding egg and albumen/white weight. There were significant differences between March, April, May and June for almost all the variables measured except for shell^L*^ (between March–May and April–June), shell^b*^ (among any of the months compared), white height (between March and May themselves and between the months of March and May and April), Yolk color (March-June) and Yolk^L*^ (between June and May themselves and between the months of June and May and April). Minor diameter, yolk^L*^, yolk^a*^, and shell weight were significantly different (*p* < 0.05) when hens and pullets were compared, with hens having a significantly higher median than pullets for all the variables except for yolk^L*^.

### 3.3. External and Internal Quality-Related Variables Canonical Correlation Analysis

The results of the canonical correlation produced three significant canonical correlations, when yolk and white pH were included and four significant correlations when such factors were not considered as shown in Table 3 and Appendix A, respectively. 

For the model that did not include pH values for yolk and white, the first, second and third significant canonical correlations produced Wilk’s Lambda values that were found to be highly significant through the use of a chi-square test that yielded *p* < 0.001. The fourth canonical correlation also proved significant at the *p* < 0.05 level. However, only the first and second canonical correlations were highly significant (*p* < 0.001) and the third canonical correlation was significant (*p* < 0.05) when yolk and white pH were not considered. All other canonical correlations were found to be non-significant.

When we did not consider yolk and white pH, the first, second, third and fourth canonical correlations produced an r (R_c_) of 0.936 which indicates that the four variates have a shared variance (r^2^ or $R_{c}^{2}$) of 87.6%, respectively (Table 3 and Appendix A). The literature proposes three methods to determine the relative importance of each original variable in each function: (1) canonical weights (standardized coefficients), (2) canonical loadings (structural correlations) and (3) canonical cross-loadings. As the canonical weights, are vulnerable to multicollinearity, the use of canonical loadings or cross-loadings is recommended (Table 4 and Appendix A). 

Significance was determined by using factor loading guidelines commonly found in the literature considering the sample sizes of 194 (when yolk and white pH were not considered) and 97 (when yolk and white pH had been included) [38,39,40,41]. We used both loadings and cross-loadings; however, there is no established cutoff. There is a rule of thumb that if any variable loading is ≥|0.30|, then it can be considered to be an important contributing variable in the function. However, this is only for explanatory studies. Hair, Black, Babin and Anderson [38] discuss the ideal case for each factor loading, i.e., the common variance should be greater than the unique one (Wilk’s Lambda ≥0.72 in order to have a variance ≥0.50), but mainly for the average; that is the reason why we use the average variance attracted (AVE ≥ 0.50). In some cases, especially a new measure, lambda ≥0.5 (AVE > 0.25) can be considered to be acceptable (but we have to address the limitation of this low AVE measure). In our case, loadings ≥|0.53| were used considering the sample size (n = 97) that included yolk and white pH among the variables, while the greater sample when both variables were excluded (n = 194) permitted considering loadings ≥|0.39| following Hair, Black, Babin and Anderson [38] criteria.

Egg weight showed a strong negative loading on the first canonical variate of external quality-related traits. Hence the first canonical variate was given the title “external lightness”, reflecting upon the negative values associated with the pool of internal quality-related variables. 

In addition, only white and yolk weight were found to have significant negative loadings on the second canonical variate of internal quality-related traits. Because of this, the first canonical variate on internal quality-related traits was given the title of “internal lightness”.

A negative loading means that eggs scoring high on the canonical variate will tend to score low on the variable, and vice versa. Hence, the heavier the egg, white and yolk weight is, the lower the score it will receive on external and internal lightness canonical covariates, respectively.

Shell^b*^ (Shell b*, shell yellow/blue coordinate) showed a strong negative loading on the second canonical variate of external quality-related traits. Hence the second canonical variate was given the title “external yellow/blue coordinate absence”, reflecting upon the negative and positive values associated with the pool of internal quality-related variables. 

Interestingly, while yolk lightness (Yolk^L*^, Yolk L*) and shell weight were found to have significant positive loadings on the second canonical variate of internal quality-related traits, yolk weight loading, significantly scored negatively. Because of this, the second canonical variate on internal quality-related traits was given the title of “internal brightness”. This means those eggs presenting a high internal brightness present high yolk lightness and shell weight values and low yolk weight.

The minor diameter showed a moderate positive loading on the third canonical variate of external quality-related traits. By contrast, shell^b*^ (Shell b*, shell yellow/blue coordinate) showed a strong negative loading on the third canonical variate of external quality-related traits. Hence the third canonical variate was given the title “egg wideness”, reflecting upon the negative and positive values associated with the pool of internal quality-related variables. Eggs with a higher wideness presented lower values for the shell yellow/blue coordinate.

Yolk^L*^ and Yolk^a*^ color decompositions (Yolk L* or lightness and Yolk a* or red/green coordinate, respectively) and yolk weight were found to have significant moderate negative loadings on the third canonical variate of internal quality-related traits. However, yolk^b*^ (Yolk b* or yellow/blue coordinate) significantly scored positively. Due to this, the third canonical variate on internal quality-related traits was given the title of “yolk dullness and yellow dominance”. This means that those eggs presenting a high yolk lightness and yellowness present high values for the yolk yellow/blue coordinate and low values for the yolk lightness color coordinate, yolk red/green coordinate and yolk weight.

The minor diameter, Shell^L*^ (Shell L*, shell lightness) and Shell^b*^ (Shell yellow/blue coordinate) showed a high negative loading on the fourth canonical variate of external quality-related traits. By contrast, the major diameter showed a moderate positive loading on the fourth canonical variate of external quality-related traits. Hence, the fourth canonical variate was given the title “egg length and external dullness”, reflecting upon the negative and positive values associated with the pool of internal quality-related variables. This means that the eggs which were longer were also duller and reported lower values for the shell yellow/blue coordinate, thus they were orangish.

The Yolk^L*^, Yolk^a*^ and Yolk^b*^ color lightness and coordinates (Yolk L* or lightness, Yolk a* or red/green coordinate, and Yolk b* or yellow/blue coordinate, respectively) and yolk weight were found to have significant moderate positive loadings on the fourth canonical variate of internal quality-related traits with Yolk^L*^ or Yolk lightness reporting the highest loading (0.670). However, Shell weight and yolk color measured with the DSM Yolk Color Fan (formerly Roche Yolk Color Fan) significantly scored negatively. Due to this, the fourth canonical variate on internal quality-related traits was given the title of “yolk yellowness and lightness, color coordinates balance, shell lightness”. This means those eggs presenting high yolk lightness and scoring low in the DSM Yolk Color Fan (yellowish) present balanced values for the yolk yellow/blue and yolk red/green coordinates and low shell weights. 

Table 5 and Appendix A show the imbalance between the proportion of variance explained by each of the canonical variates of the two sets of variables (external and internal quality-related traits) and their opposites. The proportion of the variance of external quality-related variables explained by its own canonical variate (38.2% to 35.1%, when yolk and white pH were included and excluded respectively) was slightly different to the proportion of variance of internal quality-related variables explained by opposite canonical variate (35.5% to 28.5% when yolk and white pH were included and excluded respectively). By contrast, the proportion of variance of external quality-related variables explained by its own canonical variate (15.6% to 8.0%, when yolk and white pH were included and excluded respectively) was similar to the proportion of variance of internal quality-related variables explained by opposite canonical variate (14.5% to 11.2% when yolk and white pH were included and excluded respectively).

## 4. Discussion

The demand for products deriving from non-industrial production systems has triggered and increased the interest in more sustainable farming practices, enabling the introduction of products stemming from native breeds in the common production systems and commercial chains [14]. This context lays the basis for the characterization of the quality of differentiated products linked to sustainable production involving autochthonous breeds. The differences in the values obtained in this study for egg quality-related parameters may promote the definition of products depending on which and at which level egg components are present across the different varieties and breeds studied. For instance, eggs with greater egg yolk proportions may make for richer and softer baked final products and better quality pasta, while egg whites provide the resulting products with lighter and airier textures and are richer in lysozyme [41], which is currently, the only lysozyme industrially applied for food applications.

Among external egg quality-related traits, the Leghorn hen breed’s eggs were heavier than those from the Utrerana hen breed, due to the higher weight of their shell and white. By contrast, although the Franciscan variety presented eggs with lower weight, the eggs of the Utrerana breed generally presented similar or heavier weights than other Spanish breeds [42,43,44].

The eggs of laying pullets presented a significantly lower weight than the eggs of laying hens. In addition, egg weight was observed to increase with the age of the flock hens (in consecutive months), except for March, when a higher weight of eggs was observed in the flock than that reported in April or May. The fact that first laying hens had not yet started laying eggs in March may be one of the main reasons for this finding. These results are supported by other studies in which hens of different laying periods were compared [45,46,47]. Some authors report a simultaneous increase in egg weight while there is a decrease in shell weight, which may be attributable to such parameters being conditioned by the weight of egg components (yolk and albumen). Simultaneously, egg weight has been reported to increase as the age of hens increases, while eggshell quality deteriorates, which translates in greater quality larger chicks [46].

The major diameter of the eggs was often related to the weight of the eggs. As results showed, the Leghorn eggs had significantly longer major diameters and minor diameters. However, the partridge variety reached the same major diameter as the Leghorn eggs. In addition, as previously described by Saatci et al. [46], a smaller size of the eggs of the first laying hens was observed. Variety or plumage color has been reported to significantly affect egg weight in other local bird breeds such as in Native Turkish Geese (*p* < 0.05). However, such differences were not observed regarding shape index (*p* > 0.05), or length or breadth (parameters involved in the calculation of shape index) as opposed to our results [45].

Another important characteristic of the commercialization of the product is the eggshell color profile that represents an important trait for consumer’s perception. Almost equal numbers of brown and white eggs are sold in the markets of some countries such as Spain, Germany, and Holland [48]. In the present study, a significant increase in lightness (Shell^L*^ values) on Leghorn eggs regarding Utrerana breed was observed. However, in terms of redness (Shell^a*^ values) and yellowness (Shell^b*^ values), the Utrerana breed showed higher values. These results could be due to a large amount of genetic variation for eggshell characteristics [49]. 

An increased value of shell^a*^ was observed in the Franciscan variety, suggesting the hybridization with the Plymouth Rock breed (a breed with barred feathers and brown eggs), which was carried out while aiming at defining the barred feather characteristic in the Utrerana, also added to the appearance of the undesirable characteristic of darker shell eggs. No significant differences were observed to shell color between the white variety of Utrerana and Leghorn. When Shell^L*^ value was considered to measure for eggshell lightness [50], both the white Utrerana variety and Leghorn breed reported the brightest shell tone of all remaining varieties studies.

According to other authors, the month of laying did not have a significant effect on shell weight; although egg size increases with the hen’s age, the shell weight mantains values around the same range [45,51,52]. Heat stress reduces the shell thickness and the shell quality in laying hens [53,54]. However, the Utrerana eggshell weight showed no significant differences in all the studied months. It is well known that the south of Spain is influenced by Mediterranean weather—maximum temperatures of 40 °C were reached in June 2018 in Cordoba, as reported by the State Meteorological Agency (AEMET) from Spain, with very high temperatures since late spring and summer. Taking into account that this study occurred during this period, these results suggest that the Utrerana breed tolerates high temperature-induced stress, so this might be an interesting alternative to commercial production systems with fewer adapted animals.

The Utrerana breed showed a lower eggshell weight in comparison with the Leghorn breed. Modern commercial birds showed clear differences in terms of shell weights in regard to traditional breeds [49,55]. Nevertheless, the selection of breeds for one characteristic such as egg weight can affect others such as the quality of the eggshell [56]. By contrast, Sreenivas et al. [57] suggested that Leghorn eggshell contributes a lower proportion to overall egg weight when compared to native poultry.

Some authors have reported the characteristics of the egg white to be conditioned by the strain of bird and genetic selection [58,59,60]. In this study, the Leghorn white weight was significantly heavier than those of the Utrerana varieties. In the Franciscan variety, the white weight was significantly lower than in the rest of the varieties. This could explain the lower weight of the eggs of this variety. No significant differences were observed in white weight in all the studied months, neither between the laying pullets, nor the laying hens. However, there were significant differences between June and the rest of the months, which suggests that the white height reduces as age increases, supported by the findings of Renden et al. [61].

Although the yolk diameter did not differ between breeds, the yolk weight was significantly higher in the Utrerana breed. The selection of the modern lines of laying hens induced an increase in egg weight, which translated into a simultaneous decrease in the energy content of the egg as a direct consequence of a decrease in the percentage of egg yolk. The egg white contains a larger amount of water than the yolk which results in heavier eggs. This greater contribution to egg weight is produced at a lower energetic cost as its synthesis is energetically more efficient, on a weight for weight basis, than deposition of yolk which contains proportionally 0.5 of solids with equal proportions of fat and protein [24,49]. 

At the same time, as consumers begin to demand and consider egg energy as a quality criterion, egg selection for a higher percentage of yolk will be necessary [62]. The yolk weight of the partridge and Franciscan varieties was significantly higher than the rest of the varieties or even the Leghorn breed, which may make them profitable alternatives.

An important aspect for the commercialization of the eggs is yolk color as European consumers tend to prefer darker ones, given the psychological misattribution of a healthier origin [14]. The strain of the laying hen has been suggested to determine egg pigmentation [63]. In addition, the yolk darkness is determined by the Yolk^a*^ value [50]. In the present study, yolk^a*^ and yolk^b*^ values were observed to be higher in Utrerana eggs than in Leghorn ones. On the other hand, in April and May, the yolk^a*^ value decreased, in comparison to what happened in March and June. This finding suggests that yolk^a*^ value, which accounts for the darker yolk in Utrerana, significantly decreased when the laying of the hen was higher. This suggests that the reduction in the yolk^a*^ color coordinate could be a consequence of the dilution effect originated by the increase in egg production [49]. 

A higher Yolk^a*^ value was found for the Franciscan variety, which could be linked to the higher Shell^a*^ value observed for the same variety. However, Aygun [50] reported that there was no significant correlation between eggshell color and that of the yolk. Besides, no significant differences were detected between the color of the eggshell and yolk color between white variety and Leghorn, suggesting that there was a great resemblance in the egg color of the two breeds when their plumage was white.

White quality is affected by the age of the laying hens, the strain of the birds and the storage time of the eggs [58]. The Leghorn hen’s eggs showed higher values of white height than the Utrerana breed’s eggs. The Leghorn breed laid heavier eggs with a higher proportion of white too. These results could suggest that the white height is correlated with the percentage of white, in accordance with similar observations in earlier reports [57]. When months were compared, the white height presented lower values in June, when the temperature increased. During the storage of eggs, a decrease in white height at higher temperatures has been reported by Keener et al. [64].

The Utrerana breed’s eggs reported higher white pH values in comparison to the Leghorn breed’s eggs. The white pH has been suggested to increase as CO_2_ decreases inside the egg. Factors related to this loss of CO_2_ such as the time of storage and high temperatures have been suggested to promote such a pH increase and a subsequent decrease in white viscosity and flavor, hence directly depreciating egg quality [47].

According to our results for the canonical correlation analysis (Table 5 and Appendix A), the higher the value an egg scores on the external lightness variable of egg weight, the more likely this egg will also score a higher value on internal lightness variables such as white and yolk weight, as it was also reported for the same phenotypical correlations in Japanese quails, exotic Isa brown layers and naked neck, normal and dwarf strains of Tswana chickens [65,66,67].

Similarly, those scoring a low value on the external yellow/blue coordinate are more likely to report higher values for internal brightness. Those eggs presenting high egg wideness presented higher yolk dullness and yolk yellow dominance. By contrast, the longer and duller the egg was externally, the more yellowish and lighter it was, the more balanced their color coordinates and the less heavy their shells. These results contradict some previous studies in which a weaker statistical analysis is performed [50,68], and in which it is stated that the external color of the egg is not related to its internal color. Given our results, the Utrerana breed eggs allow consumers to associate the external appearance of the egg with their internal characteristics. The relationship between the outer shape of the egg and the internal color of the egg may suggest that there could be a dilution effect of pigments depending on the shape and size of the egg as reported by other studies [49].

The moderately high values for the proportion of variance of external quality-related variables explained by its own and opposite canonical variable, which doubles the explanatory power of variance of the internal quality-related traits set, suggest external quality-related traits may have a remarkably 2-fold higher predictive power of internal quality-related traits than vice versa. Interestingly, the reduction in the proportion of variance of internal quality-related variables, explained by its own canonical variate from 15.6 to 8.0% when pH was excluded, suggests these variables (yolk and white pH) may be relevant traits to consider for the determination of internal egg quality. Furthermore, external egg quality-related traits may act as better predictors of internal quality-related traits, which is desirable as it permits not having to break the eggs to classify them, relying on their internal quality to enable the implementation of an effective noninvasive method for internal quality determination.

## 5. Conclusions

Involving autochthonous breeds in common production systems and commercial chains seeking the characterization of the quality of differentiated products could be the key to future poultry sustainable productions. Leghorn eggs are heavier than those from Utrerana hens; however, these generally presented similar or heavier weights than other Spanish breeds. There is a simultaneous increase in egg weight and a decrease in shell weight, which may be conditioned by the weight of egg components (yolk and albumen). Egg weight increases with age, while eggshell quality deteriorates. The variety or plumage color affects egg weight and egg length or breadth. Utrerana hybridization with the Plymouth Rock breed (a breed with barred feathers and brown eggs) added to the appearance of the undesirable characteristic of darker shell eggs, while the possible hybridization between the white Utrerana variety and Leghorn breed may account for the increased values for shell brightness reported. The Utrerana breed may tolerate high temperature-induced stress better than the Leghorn breed, so this might be an interesting alternative to commercial production systems with fewer adapted animals. The Leghorn breed’s white weight was significantly heavier than those of the Utrerana varieties. The white height reduces as age increases. The modern line selection of laying hens has induced an increase in egg weight, which translates into a simultaneous decrease in the energy content of the egg, as a direct consequence of a decrease in the percentage of egg yolk. As white pH increases, CO_2_ content decreases inside the egg. Simultaneous to this decrease, there is a subsequent decrease in white viscosity and flavor, which directly depreciates egg quality. The canonical correlation analysis addresses the possibility to develop a tool comprising external indicators that may indirectly report information on certain determinants of the internal quality of these eggs. This could mean a great advancement in the identification and typification of specific products, which may cover the currently increasing demand from markets for non-conventional quality products linked to specific breeds or production systems, or even settle new commercialization niches linked to local defined traceable products.

## Figures and Tables

**Table 1 animals-09-00153-t001:** Flock management information. All cages were chosen according to Council Directive 1999/74/EC of 19 July, 1999, laying down minimum standards for the protection of laying hens.

Flock Management Parameter	Utrerana				Leghorn (Control)
	White	Franciscan	Black	Partridge	
Breeding hens	17	17	17	17	17
Hens (70 weeks old)	12	12	12	12	12
Pullets (28 weeks old)	5	5	5	5	5
Stocking density	4 animals per each m^2^				
Nest box density	29 animals per each m^2^				
Waterer allotment/space	Circle waterers of 5 cm of diameter per animal				
Feeder allotment/space	41 cm per animal				
Floor substrate	Wood shavings covering the floor at a depth of approximately 1 cm				
Nest box substrate	Plastic turf mats covering the floor at a depth of approximately 1 cm				

**Table 2 animals-09-00153-t002:** Summary of the results for the Kruskal–Wallis H test and the determinative coefficient through r or partial eta squared (ηp^2^), for the fixed effects for internal and external egg quality traits from the model, excluding yolk and white pH in Utrerana hens (n = 194).

Variable	Parameter	Egg Weight	Major Diameter	Minor Diameter	Shell^L*^	Shell^a*^	Shell^b*^	White Height	Yolk Color	Yolk^L*^	Yolk^a*^	Yolk^b*^	Yolk Diameter	Shell Weight	Yolk Weight	White Weight
Month	χ^2^	5.156	6.423	1.139	15.831	1.415	10.753	8.854	52.954	67.718	82.856	89.812	1.056	7.675	3.071	4.411
	dfn	3	3	3	3	3	3	3	3	3	3	3	3	3	3	3
	*p*-value	0.161	0.093	0.768	0.001	0.702	0.013	0.031	0.000	0.000	0.000	0.000	0.788	0.053	0.381	0.220
	dfd	192	192	192	192	192	192	192	192	192	192	192	192	192	192	192
	F	1.719	2.141	0.380	5.277	0.472	3.584	2.951	17.651	22.573	27.619	29.937	0.352	2.558	1.024	1.470
	ηp^2^	0.026	0.032	0.006	0.076	0.007	0.053	0.044	0.216	0.261	0.301	0.319	0.005	0.038	0.016	0.022
Order	χ^2^	13.672	8.945	10.682	5.844	3.418	1.777	9.268	2.178	1.245	1.702	6.035	4.371	7.241	4.262	25.574
	dfn	5	5	5	5	5	5	5	5	5	5	5	5	5	5	5
	*p*-value	0.018	0.111	0.058	0.322	0.636	0.879	0.099	0.824	0.940	0.889	0.303	0.497	0.203	0.512	0.000
	dfd	190	190	190	190	190	190	190	190	190	190	190	190	190	190	190
	F	2.734	1.789	2.136	1.169	0.684	0.355	1.854	0.436	0.249	0.340	1.207	0.874	1.448	0.852	5.115
	ηp^2^	0.067	0.045	0.053	0.030	0.018	0.009	0.047	0.011	0.007	0.009	0.031	0.022	0.037	0.022	0.119
Period	χ^2^	1.366	2.360	0.350	1.211	0.022	5.345	7.700	6.149	10.135	11.999	4.589	2.684	7.185	3.296	0.422
	dfn	2	2	2	2	2	2	2	2	2	2	2	2	2	2	2
	*p*-value	0.505	0.307	0.840	0.546	0.989	0.069	0.021	0.046	0.006	0.002	0.101	0.261	0.028	0.192	0.810
	dfd	193	193	193	193	193	193	193	193	193	193	193	193	193	193	193
	F	0.683	1.180	0.175	0.606	0.011	2.673	3.850	3.075	5.068	6.000	2.295	1.342	3.593	1.648	0.211
	ηp^2^	0.007	0.012	0.002	0.006	0.000	0.027	0.038	0.031	0.050	0.059	0.023	0.014	0.036	0.017	0.002
Laying Age	χ^2^	3.666	1.657	4.491	3.065	0.15	0.291	1.078	2.879	6.163	4.382	0.831	0.379	11.707	0.323	3.754
	dfn	1	1	1	1	1	1	1	1	1	1	1	1	1	1	1
	*p*-value	0.056	0.198	0.034	0.08	0.699	0.590	0.299	0.09	0.013	0.036	0.362	0.538	0.001	0.57	0.053
	dfd	194	194	194	194	194	194	194	194	194	194	194	194	194	194	194
	F	3.666	1.657	4.491	3.065	0.150	0.291	1.078	2.879	6.163	4.382	0.831	0.379	11.707	0.323	3.754
	r	0.019	0.008	0.023	0.016	0.001	0.001	0.006	0.015	0.031	0.022	0.004	0.002	0.057	0.002	0.019
Variety	χ^2^	34.300	39.881	29.270	36.567	65.585	98.046	15.421	15.238	5.696	16.295	15.891	21.296	62.090	54.267	59.063
	dfn	4	4	4	4	4	4	4	4	4	4	4	4	4	4	4
	*p*-value	0.000	0.000	0.000	0.000	0.000	0.000	0.004	0.004	0.223	0.003	0.003	0.000	0.000	0.000	0.000
	dfd	191	191	191	191	191	191	191	191	191	191	191	191	191	191	191
	F	8.575	9.970	7.318	9.142	16.396	24.512	3.855	3.810	1.424	4.074	3.973	5.324	15.523	13.567	14.766
	ηp^2^	0.152	0.173	0.133	0.161	0.256	0.339	0.075	0.074	0.029	0.079	0.077	0.100	0.245	0.221	0.236
Breed	χ2	22.546	8.765	21.607	31.945	49.3	92.019	11.455	8.597	0.873	7.845	10.748	3.502	55.027	23.651	29.992
	dfn	1	1	1	1	1	1	1	1	1	1	1	1	1	1	1
	*p*-value	0.000	0.003	0.000	0.000	0.000	0.000	0.001	0.003	0.350	0.005	0.001	0.061	0.000	0.000	0.000
	dfd	194	194	194	194	194	194	194	194	194	194	194	194	194	194	194
	F	22.546	8.765	21.607	31.945	49.300	92.019	11.455	8.597	0.873	7.845	10.748	3.502	55.027	23.651	29.992
	r	0.104	0.043	0.100	0.141	0.203	0.322	0.056	0.042	0.004	0.039	0.052	0.018	0.221	0.109	0.134

χ^2^: Chi squared; dfn: degrees of freedom numerator; dfd: degrees of freedom denominator. ηp^2^ can be benchmarked against the Cohen [26] criteria of small (0.01), medium (0.09), and large (0.25) effects as suggested in Richardson [27]. In Cohen’s terminology, a small effect size is one in which there is a real effect but which you can only see through careful study. By contrast, a ‘large’ effect size is an effect which is big enough, and/or consistent enough, that you may be able to see it ‘with the naked eye’. Cohen’s guidelines for *r* are that a small effect is 0.1, a medium effect is 0.3, and a large effect is 0.5.

**Table 3 animals-09-00153-t003:** Standardized canonical coefficients of variables, canonical correlations between two sets of variables (r), squared canonical correlation (r^2^) and their probabilities (F) for internal and external egg quality-related traits, excluding yolk and white pH, in Utrerana hens compared to laying lineage (n = 194).

Canonical Pairs	1st	2nd	3rd	4th	5th	6th
r (R_c_)	0.936	0.616	0.394	0.314	0.260	0.068
r^2^ $\left(R_{c}^{2}\right)$	0.876	0.379	0.155	0.099	0.068	0.05
F	12.999	4.071	2.348	1.856	1.382	0.214
Degrees of Freedom	54	40	28	18	10	4
Sig.	0.000	0.000	0.000	0.017	0.187	0.931
Standardized canonical coefficients of external quality-related traits						
Egg weight	**−0.972**	0.139	−0.065	0.240	−0.934	1.888
Major diameter	0.025	−0.345	−0.337	**0.575**	0.606	−1.464
Minor diameter	−0.066	0.091	**0.520**	**−0.859**	0.597	−1.059
Shell^L*^	−0.021	0.134	−1.327	**−0.787**	0.763	0.167
Shell^a*^	−0.076	0.251	−0.322	−0.104	−1.092	−0.443
Shell^b*^	0.015	**−0.967**	**−0.929**	**−0.838**	0.993	0.545
Standardized canonical coefficients of internal quality-related traits						
White height	−0.037	0.229	−0.219	−0.311	−0.194	0.462
Yolk color	−0.030	−0.189	0.139	**−0.543**	0.224	−1.428
Yolk^L*^	−0.010	**0.616**	**−0.944**	**0.670**	0.136	−0.308
Yolk^a*^	0.040	0.198	**−1.370**	**0.449**	−0.206	1.537
Yolk^b*^	0.093	−0.031	**0.543**	**0.479**	−0.630	−1.367
Yolk diameter	−0.026	0.010	−0.09	−0.139	−0.089	0.102
Shell weight	−0.305	**0.551**	0.188	**−0.405**	−0.607	−0.595
Yolk weight	**−0.400**	**−0.470**	**−0.538**	−0.098	−0.132	0.313
White weight	**−0.702**	−0.277	0.306	**0.694**	0.446	−0.164
**Bold**: Given the sample size of 194, a criterion of ≥|0.39| was considered for variable loadings to be significant [38].						

**Table 4 animals-09-00153-t004:** Correlations between the variables and related canonical variables (canonical loadings) and between the variables and the other set of canonical variables (canonical cross-loadings) for internal and external egg quality-related traits, excluding yolk and white pH, in Utrerana hens compared to laying lineage (n = 194).

Variable	U_1_	U_2_	U_3_	U_4_	U_5_	U_6_	V_1_	V_2_	V_3_	V_4_	V_5_	V_6_
Egg weight	**−0.996**	**−0.014**	**0.003**	**0.070**	**0.046**	**0.032**	−0.932	−0.009	0.001	0.022	0.012	0.002
Major diameter	**−0.745**	**−0.211**	**−0.222**	**0.448**	**0.168**	**−0.349**	−0.697	−0.130	−0.088	0.141	0.044	−0.024
Minor diameter	**−0.742**	**0.060**	**0.347**	**−0.497**	**0.114**	**−0.255**	−0.695	0.037	0.136	−0.156	0.030	−0.017
Shell^L*^	**−0.087**	**0.817**	**−0.471**	**−0.018**	**0.319**	**0.039**	−0.081	0.503	−0.185	−0.006	0.083	0.003
Shell^a*^	**0.013**	**−0.351**	**−0.292**	**−0.362**	**−0.753**	**−0.305**	0.012	−0.216	−0.115	−0.114	−0.196	−0.021
Shell^b*^	**0.044**	**−0.934**	**−0.029**	**−0.293**	**−0.194**	**0.031**	0.041	−0.575	−0.011	−0.092	−0.050	0.002
White height	−0.364	0.177	0.063	−0.088	−0.010	0.023	**−0.389**	**0.288**	**0.160**	**−0.281**	**−0.038**	**0.344**
Yolk color	−0.039	−0.260	−0.132	−0.122	0.016	−0.034	**−0.042**	**−0.422**	**−0.335**	**−0.389**	**0.062**	**−0.505**
Yolk^L*^	−0.011	0.372	−0.157	0.066	0.126	−0.006	**−0.011**	**0.604**	**−0.399**	**0.209**	**0.486**	**−0.093**
Yolk^a*^	0.038	−0.243	−0.091	0.034	−0.149	−0.010	**0.040**	**−0.394**	**−0.232**	**0.109**	**−0.573**	**−0.145**
Yolk^b*^	0.201	−0.249	0.022	0.134	−0.189	−0.011	**0.215**	**−0.405**	**0.057**	**0.426**	**−0.728**	**−0.156**
Yolk diameter	−0.288	−0.079	−0.028	−0.027	−0.050	0.003	**−0.308**	**−0.129**	**−0.072**	**−0.087**	**−0.193**	**0.048**
Shell weight	−0.582	0.340	0.025	−0.079	−0.100	−0.009	**−0.622**	**0.553**	**0.064**	**−0.250**	**−0.384**	**−0.132**
Yolk weight	−0.390	−0.313	−0.194	−0.044	−0.037	0.003	**−0.417**	**−0.508**	**−0.492**	**−0.139**	**−0.143**	**0.051**
White weight	−0.800	−0.015	0.102	0.082	0.039	−0.001	**−0.855**	**−0.024**	**0.260**	**0.262**	**0.151**	**−0.010**

U1, U2, U3, U4, U5, U6: canonical variates containing external quality-related traits; V1, V2, V3, V4, V5, V6: canonical variates containing internal quality-related traits. **Bold**: canonical loadings; Regular: canonical cross-loadings.

**Table 5 animals-09-00153-t005:** Proportion of explained variance, eigenvalues and percentages of explained common variance associated with each factor of internal and external egg quality-related trait, excluding yolk and white pH, in Utrerana hens compared to laying lineage (n = 194).

Canonical Variable	1st	2nd	3rd	4th	5th	6th
Eigenvalue	7.056	0.610	0.184	0.110	0.072	0.05
Wilk’s Λ Statistic	0.054	0.439	0.707	0.836	0.928	0.995
Proportion of variance of external quality-related variables explained by its own canonical variate $\left(U_{e}^{2}\right)$	0.351	0.308	0.176	0.154	0.351	0.308
Proportion of variance of external quality-related variables explained by opposite canonical variate $\left(V_{e}^{2}\right)$	0.285	0.108	0.169	0.064	0.285	0.108
Proportion of variance of internal quality-related variables explained by its own canonical variate $\left(U_{i}^{2}\right)$	0.080	0.012	0.075	0.012	0.080	0.012
Proportion of variance of external quality-related variables explained by opposite canonical variate $\left(V_{i}^{2}\right)$	0.112	0.011	0.069	0.007	0.112	0.011

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
