# Peer review of "Non-Parametrical Canonical Analysis of Quality-Related Characteristics of Eggs of Different Varieties of Native Hens Compared to Laying Lineage"

_animals, 2019, doi:10.3390/ani9040153_

Reviewer 1 Report

L28: In the abstract, the authors should include some key factors and examples of levels detected for various factors monitored; etc. 

L102: The authors provide the following minimum flock management information (this could be summarized in a table): breeding hens; hen age; stocking density; nest box density; waterer allotment/space; floor substrate; nest box substrate.  The type of cages utilized should also be thoroughly described since in world-wide audience will have various assumptions of conventional and enriched colony options.

L128: More information should be provided.  At minimum, the authors must supply the instrumentation and sources utilized, as well as the number of eggs assessed per sample period, the number of sample periods, and egg temperature at the time of egg quality assessment (since egg quality measurements are temperature dependent).  Additionally, were eggs pooled for all or some analyses? 

Author Response

All the team responsible for this paper acknowledge the comments from the reviewer and editor, as they help to improve the quality of our manuscript. In the following paragraphs, we will describe and address how the referee’ recommendations were followed.

L28: In the abstract, the authors should include some key factors and examples of levels detected for various factors monitored; etc. 

Response: We included further information regarding the differences found among the different levels for the variables assessed. We rearranged the abstract after the information was included in order to make it fit to the limit of words.

L102: The authors provide the following minimum flock management information (this could be summarized in a table): breeding hens; hen age; stocking density; nest box density; waterer allotment/space; floor substrate; nest box substrate.  The type of cages utilized should also be thoroughly described since in world-wide audience will have various assumptions of conventional and enriched colony options.

Response: table 1 was included according to reviewer’s suggestion.

L128: More information should be provided.  At minimum, the authors must supply the instrumentation and sources utilized, as well as the number of eggs assessed per sample period, the number of sample periods, and egg temperature at the time of egg quality assessment (since egg quality measurements are temperature dependent).  Additionally, were eggs pooled for all or some analyses? 

Response: Lines 147 to 150. The information required by reviewer was provided. Eggs were not pooled for any of the analyses in this study.

Reviewer 2 Report

The manuscript refers about a very accurate statistical analysis of several quality traits of eggs laid by some varieties of Utrerana hens at different ages.

Introduction is appropriate, materials and methods are redundant whit particular regards to the  theory of the statistical tools used to analyse the data. The expalined concepts are well known theoretical bases of non parametric and canonical correlation analysis. I suggest to summarize this part. Table 1 showing the results of the normality test is not necessary (it is enough to report them in the 'results section' as you already done).

In the results section, it will be interesting to add some sentences about the results of Mann-Whitney tests and Dunn test pairwise comparison (decrements or increments, in percentage, passing between the used variable manipulation).

Discussion can be improved since some parts did not provide sufficient explanations to the results obtained in this study.

Conclusions can be improved by linking them more to the results obtained.

Minor remarks:

R125 ...Yolk*L, Yolk*a, Yolk*b.. put the * after the letter and check it throughout the manuscript;

R349 ....(YolkL*, Yolk L*) please check it;

R406-409 not clear, please reformulate;

R414 check 'where'

R427-430 not clear please reformulate;

R470 please check,

Author Response

All the team responsible for this paper acknowledge the comments from the reviewer and editor, as they help to improve the quality of our manuscript. In the following paragraphs, we will describe and address how the referee’ recommendations were followed.

The manuscript refers about a very accurate statistical analysis of several quality traits of eggs laid by some varieties of Utrerana hens at different ages.

Introduction is appropriate, materials and methods are redundant whit particular regards to the  theory of the statistical tools used to analyse the data. The expalined concepts are well known theoretical bases of non parametric and canonical correlation analysis. I suggest to summarize this part. Table 1 showing the results of the normality test is not necessary (it is enough to report them in the 'results section' as you already done).

Response: We summarized the statistical analysis section and have reduced it 422 words.

In the results section, it will be interesting to add some sentences about the results of Mann-Whitney tests and Dunn test pairwise comparison (decrements or increments, in percentage, passing between the used variable manipulation).

Response: A new paragraph was added from lines 316 to lines 327 reporting summarized information regarding Mann-Whitney tests, Dunn test pairwise comparisons and independent-median tests.

Discussion can be improved since some parts did not provide sufficient explanations to the results obtained in this study.

Response: Further explanations were provided across all discussion to reinforce the statements made and the results reported. Six new citations were added to support the results included.

Conclusions can be improved by linking them more to the results obtained.

Response: Conclusions were rewritten including the results commented and discussed at the discussion section.

Minor remarks:

R125 ...Yolk*L, Yolk*a, Yolk*b.. put the * after the letter and check it throughout the manuscript;

Response: All manuscript was checked regarding YolkL*, Yolka* and Yolkb* following reviewer’s suggestion.

R349 ....(YolkL*, Yolk L*) please check it;

Response: All manuscript was checked regarding YolkL*, Yolka* and Yolkb* following reviewer’s suggestion.

R406-409 not clear, please reformulate;

Response: Lines 440-445, The content was reformulated as suggested.

R414 check 'where'

Response: Line 452. corrected.

R427-430 not clear please reformulate;

Response: Lines 465 to 469. Content was reformulated according to reviewer’s suggestion.

R470 please check,

Response: Lines 503 to 510 were reformulated.

Reviewer 3 Report

The authors are comparing the external and internal quality of eggs laid by a commercial strain of hen with those from four different colour types of an indigenous breed.  The manuscript focusses mainly on the extremely detailed and complex statistical analysis of the data obtained.  No where in the manuscript is there a simple summary of the means of the data for each of the variables measured - in my opinion this needs to be added.  My expectation is that the readers of the journal will be more interested in the basic results obtained from the two breeds and less interested in the detailed statistical analysis.

I have suggested minor changes in the attached file which contains only pages that I have annotated.

Author Response

The authors are comparing the external and internal quality of eggs laid by a commercial strain of hen with those from four different colour types of an indigenous breed.  The manuscript focusses mainly on the extremely detailed and complex statistical analysis of the data obtained.  No where in the manuscript is there a simple summary of the means of the data for each of the variables measured - in my opinion this needs to be added.  My expectation is that the readers of the journal will be more interested in the basic results obtained from the two breeds and less interested in the detailed statistical analysis.

Response: Line 246. A summary of the descriptive statistics had been already included for both sets, including and excluding yolk and white pH in the analysis as Supplementary Table S6.

I have suggested minor changes in the attached file which contains only pages that I have annotated.

English language was checked by a Cambridge ESOL examinations instructor to correct for grammar and agreement issues to improve the readability of the manuscript.

Response to pdf file.

Line 18. We accepted the changes suggested by reviewer in the pdf file attached.

Line 23. We accepted the changes suggested by reviewer in the pdf file attached.

Abstract. We included all suggestions made by reviewer.

Line 54. We accepted the changes suggested by reviewer in the pdf file attached.

Line 62. We accepted the changes suggested by reviewer in the pdf file attached.

Lines 66-67. We accepted the changes suggested by reviewer in the pdf file attached.

Line 80. We accepted the changes suggested by reviewer in the pdf file attached.

Line 84. We accepted the changes suggested by reviewer in the pdf file attached.

Line 85. We accepted the changes suggested by reviewer in the pdf file attached.

Line 86. We accepted the changes suggested by reviewer in the pdf file attached.

Line 93. We accepted the changes suggested by reviewer in the pdf file attached.

Line 98. We accepted the changes suggested by reviewer in the pdf file attached.

Lines 100-101. We accepted the changes suggested by reviewer in the pdf file attached.

Lines 136, 139 and 141. We accepted the changes suggested by reviewer in the pdf file attached.

Lines 146-147. We accepted the changes suggested by reviewer in the pdf file attached.

Lines 157-158. We accepted the changes suggested by reviewer in the pdf file attached.

Lines 160-161. We accepted the changes suggested by reviewer in the pdf file attached.

Line 169. We accepted the changes suggested by reviewer in the pdf file attached.

Line 171. We accepted the changes suggested by reviewer in the pdf file attached.

Line 428. We accepted the changes suggested by reviewer in the pdf file attached.

Line 430. We accepted the changes suggested by reviewer in the pdf file attached.

Line 432. We accepted the changes suggested by reviewer in the pdf file attached.

Line 434. We accepted the changes suggested by reviewer in the pdf file attached.

Line 435. We accepted the changes suggested by reviewer in the pdf file attached.

Line 437. We accepted the changes suggested by reviewer in the pdf file attached.